# Impact of Aboveground Vegetation on Abundance, Diversity, and Biomass of Earthworms in Selected Land Use Systems as a Model of Synchrony between Aboveground and Belowground Habitats in Mid-Himalaya, India

Kasahun Gudeta [1,2,*], Ankeet Bhagat [3], Jatinder Mohan Julka [1], Sartaj Ahmad Bhat [4,*], Gopal Krishan Sharma [5], Getachew Bantihun [2], Ryszard Amarowicz [6] and Merga Belina [7]

1 School of Biological and Environmental Sciences, Shoolini University of Biotechnology and Management Sciences, Solan 173229, India
2 Department of Applied Biology, Adama Science and Technology University, Adama 1888, Ethiopia
3 Department of Zoology, Guru Nanak Dev University, Amritsar 143005, India
4 River Basin Research Center, Gifu University, 1-1 Yanagido, Gifu 501-1193, Japan
5 School of Agricultural Sciences, Shoolini University of Biotechnology and Management Sciences, Solan 173229, India
6 Institute of Animal Reproduction and Food Research, Polish Academy of Sciences, 10-748 Olsztyn, Poland
7 Department of Statistics, College of Natural and Computational Sciences, Addis Ababa University, Addis Ababa 1000, Ethiopia
* Correspondence: kggutema@shooliniuniversity.com (K.G.); sartajbhat88@gmail.com (S.A.B.)

**Abstract:** The population status and biomass of earthworms were studied in three different land use systems of pasture (Pa), silvopasture (SP), and mixed evergreen forest (MEF) from 2019–2020 in the Solan district of Himachal Pradesh, India. The aim of this study was to assess the population status of earthworms and investigate how different land use systems influence their abundance, diversity, and biomass. Earthworms and soil were sampled using the Tropical Soil Biology and Fertility (TSBF) method in all seasons (winter, spring, summer, monsoon, and autumn). The physicochemical properties of the soil were analyzed to evaluate their effects on the diversity, biomass, and density of animals. The diversity status parameters, such as the Shannon diversity index (H′), Margalef richness index (R), evenness (J′), and dominance index (D), were computed. A total of seven earthworm species, belonging to four families, namely, *Amynthas corticis*, *Aporrectodea rosea*, *Drawida japonica*, *Eisenia fetida*, *Metaphire birmanica*, *Metaphire houlleti*, and *Lennogaster pusillus*, were identified from all three land use systems. The lowest Shannon diversity index (H′), Margalef index (R), and evenness (J′) index values were registered in MEF (H′ = 0.661, R = 0.762, J′ = 0.369) compared to those in Pa (H′ = 1.25, R = 1.165, J′ = 0.696) and SP (H′ = 0.99, R = 0.883, J′ = 0.552), implying that MEF is the least diversified land system. In contrast, the highest dominance index (D) value was registered in MEF (Pa = 0.39, SP = 0.53, MEF = 0.67), which again showed that MEF is the least diversified land system. The highest values of abundance and biomass were recorded in MEF (754.15 individuals $m^{-2}$ and 156.02 g $m^{-2}$), followed by SP (306.13 individuals $m^{-2}$ and 124.84 g $m^{-2}$) and Pa (77.87 individuals $m^{-2}$ and 31.82 g $m^{-2}$). Both the density and biomass of earthworms increased from Pa to MEF (Pa < SP < MEF). This study is novel because it revealed that the diversity and productivity (biomass and abundance) values of earthworms were negatively correlated (as diversity increased, productivity decreased; as diversity decreased, productivity increased). The total values of abundance and biomass of earthworms in the three land use systems indicated perfect synchrony between aboveground and belowground habitats, whereas the diversity values revealed that MEF was dominated by one or two species and the least diversified. Therefore, for sustainable belowground productivity, aboveground conservation is recommended, and vice versa, regardless of diversity.

**Keywords:** biomass; density; diversity; earthworms; mixed evergreen forest; pasture; silvopasture

## 1. Introduction

Land use change greatly influences diversity, distribution, and abundances of earthworms [1–3]. The effect of the plant community on underground soil faunal diversity remains hazy. As a result, understanding the relations among aboveground vegetation and living organisms in soil is of significant interest. The dynamics of earthworm populations in pineapple agroecosystems were evaluated to assess the effect of monocultures on earthworm communities [4]. They found that the community of earthworms in pineapple agroecosystems predominantly consisted of endogeic earthworms. Among all the endogeic species, *Drawida assamensis* was more dominant with respect to its biomass, density, and relative abundance. In addition, ref. [5] studied the population dynamics of earthworms and found that different land use systems greatly influenced species richness, community organization, abundance, and distribution.

The community structure of both fungi and earthworms was evaluated in four different land use systems, and their community structure was found to change in an increasing manner from conventional arable land to woodland (conventional arable land → no or reduced tillage → grassland → wooded land) [6]. The density and biomass of earthworm populations were examined in a wide range of land use types, such as moderately degraded natural forests, highly degraded natural forests, rehabilitated forest lands, traditional pure crop system, and traditional agroforestry system, and abandoned and rehabilitated farmland in a village landscape of the central Himalayas, India [7]. They reported that the change from the traditional pure crop system to the traditional agroforestry system significantly increased earthworm density and biomass. Both land use intensity and land use type are strong drivers of the abundance and composition of earthworm communities in agricultural ecosystems [8]. They recorded the lowest number of earthworms in coniferous forests and intensively managed agricultural land use, an intermediate number of populations in organic no-till systems and the highest populations in ancient deciduous forest systems.

Studies have reported that changes in land use systems impact soil nutrient dynamics in Amazonia [9] and influence soil physicochemical parameters [10,11]. Earthworms are keystone species of the soil ecosystem, and their distribution, diversity, and abundance assist the total diversity, physicochemical properties, and fertility of the soil [12]. On the other hand, tree plantations affected the abundance of earthworms by altering the physicochemical properties of the soil, such as temperature, humidity, pH, and organic matter content [13,14]. It was also reported that the overall density and biomass of earthworms are higher in older forests due to the accumulation of organic matter in the area occurring through the long-term breakdown of plant and animal material [5]. Moisture has been identified to influence the diversity and abundance of earthworms that are directly affected by temperature. As the temperature increases, the moisture content decreases, which does not allow the survival, growth, and reproduction of earthworms [4]. In studies on earthworm communities in rubber plantations of different ages in West Tripura, India, the population density and total biomass of earthworms increased during the monsoon and post-monsoon seasons, which were characterized by higher humidity, which is suitable for the survival, growth, and reproduction of earthworms [15]; it was noted that both density and biomass increased, while species diversity, species richness, and species evenness decreased with increasing farm age. Compared with forest ecosystems, the diversity of functional guilds was lower in agricultural ecosystems [16].

No such studies have been carried out to reveal how aboveground vegetation influences the physicochemical parameters of soil and in turn directly affects belowground diversity. In addition, there have been no sufficient studies on the belowground diversity of the Himalayan mountain system, which is recognized as a global biodiversity hotspot [17] and requires extensive study. This study aimed to analyze the species diversity, biomass, and density of earthworms in relation to land use systems of pasture (Pa), silvopasture (SP), and mixed evergreen forest (MEF) in the Mid-Himalaya of the Solan district, India. In this study, we used earthworms (keystone species of belowground habitat) as a model to



investigate (a) how aboveground vegetation of Pa, SP, and MEF influenced belowground population status (density, biomass, and diversity) of earthworms; (b) alpha diversity and the trend of diversity indices with both biomass and density values in the land use systems. The outcomes of this research could be useful to provide insight on the trend of diversity and productivity (density and biomass) in the soil of the studied land use systems determined by habitat preference of earthworms based on adaptations that may need further investigation.

## 2. Materials and Methods

The sampling was conducted during the winter, spring, summer, monsoon, and autumn seasons at three different sites, viz., Pa, SP, and MEF land use systems, selected for this study in the same locality of a village called Chemalti, Solan, Himachal Pradesh, India (Figure 1). Solan has an average elevation of 1502 metres (5249.34 feet) at GPS coordinates of 30.9084° N, 77.0999° E and its average annual rainfall is 1262 mm. The GPS coordinates of the sampling site are 30.8628° N, 77.097° E. These different land use systems were characterized by having the following vegetation. The pastoral land use system has *Chrysopogon fulvus*, *Heteropogon contortus*, and *Apluda* spp. Silvopastoral land use systems possess *Quercus leucotricophora* (Ban Oak), shrubs, and grasses. The mixed evergreen forestland use system comprises *Quercus leucotricophora* (Ban Oak), *Pinus roxburghii* (Chir Pine), *Myrica nagi* (Kafal), *Prunus paddus* (Paja), *Celtis australis* (Khirak), *Toona ciliata* (Toon), *Ficus* spp. (Anjeer), *Indigofera pulchella* (Kathi), *Beberis asiatica* (Rasaunt/Kasgmal), *Rubus ellipticus* (Yellow Himalayan Raspberry), and *Carrisa coronda* (Coronda). The soil of the current study area is categorized as the loam soil type.

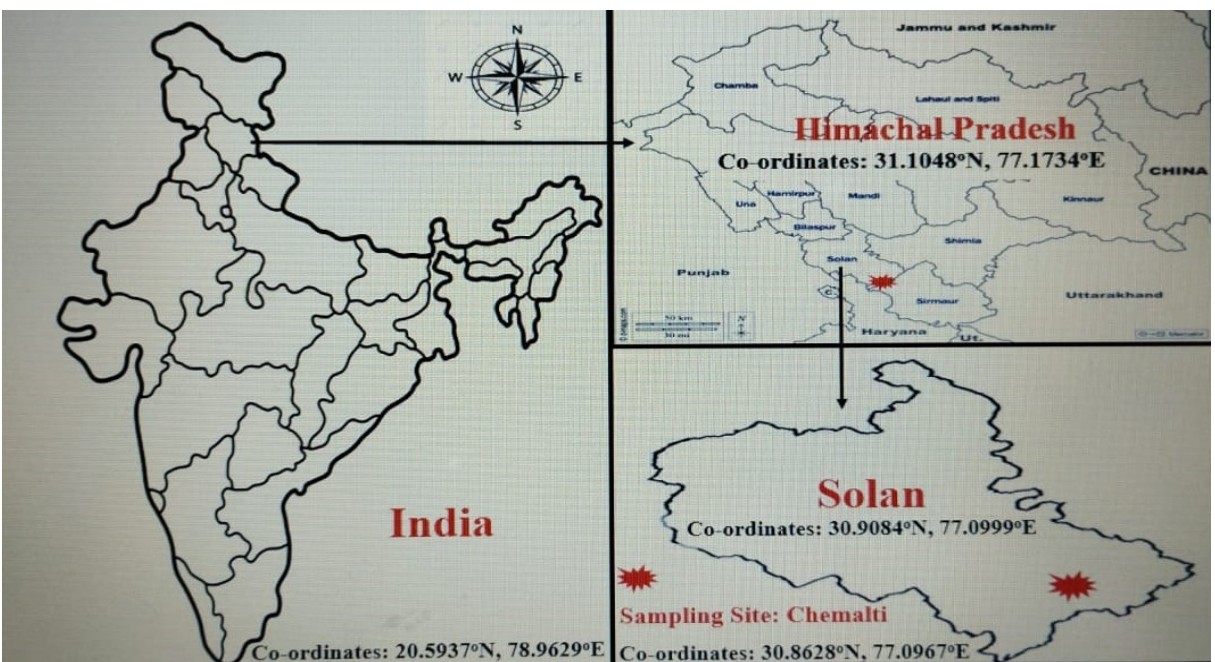

**Figure 1.** Sampling sites of the study in Solan, Himachal Pradesh, India.

### 2.1. Earthworm Sampling

Earthworms were sampled using the Tropical Soil Biology and Fertility (TSBF) method [18,19]. Soil monoliths (25 × 25 × 30 cm) were taken by digging the soil. Three spatially different plots (10 × 10 m) were marked in each land use system (Table 1). Five soil monoliths were randomly taken from each plot, resulting in a total of 15 samples from each land use type. The samples were collected for five seasons (Table 2). All samplings were carried out during different seasons over a period of two years (February 2019–

November 2020). For each land use system (Pa, SP, and MEF), triplicates were used (Pa1, Pa2, Pa3; SP1, SP2, SP3; MEF1, MEF2, and MEF3) to ensure statistically significant data.

**Table 1.** Design of sampling site allocation during each sampling time (2019–2020).

| Study Site | Plots | Area of Each Plot | No. of Monoliths from Each Plot | Total No. of Monoliths (25 × 25 × 30 cm) | Containers of Earthworm Samples |
|---|---|---|---|---|---|
| Pa | 3 | 10 m$^{-2}$ | 5 | 15 | 15 |
| SP | 3 | 10 m$^{-2}$ | 5 | 15 | 15 |
| MEF | 3 | 10 m$^{-2}$ | 5 | 15 | 15 |
| Total | 9 | | 15 | 45 | 45 |

**Table 2.** Sampling time (seasons and months) February 2019 to November 2020.

| Seasons | Months |
|---|---|
| Winter | December–February |
| Spring | March–April |
| Summer | May–June |
| Monsoon | July–September |
| Autumn | October–November |

*2.2. Identification of Earthworms*

Earthworms were hand-sorted, and the collected earthworms from the sites were labeled with the place of collection (name of site, sampling date) as shown in Figure 2. The collected earthworm samples were thoroughly washed to remove soil debris and preserved in 5% formalin to avoid color loss, as color is an important criterion for earthworm species identification. The earthworms were identified to the species level with the help of keys provided by [20–28].

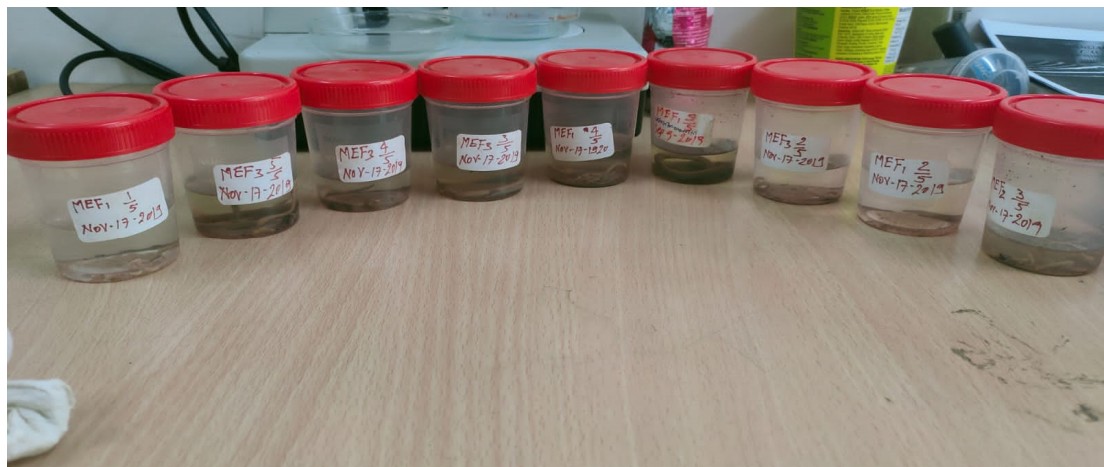

**Figure 2.** Samples of collected earthworms from different land use systems (Pa, SP, and MEF) (Photo: Kasahun Gudeta).

Reasonable amounts of soil were taken from the place where earthworms were hand-sorted in plastic bags and were labeled with the site name and the date of sampling. The physicochemical parameters of the soil samples, such as pH, temperature, moisture, and available carbon, nitrogen, phosphorous, and potassium, were analyzed using different analysis methods. The soil temperature of the different land use types in each plot was immediately measured by a soil thermometer. Soil texture was determined by the textural triangle of the soil classification system defined by the USDA [29], and moisture was detected by the gravimetric method [30]. Soil pH was measured using a digital pH meter

(Deluxe pH meter 101). Available organic carbon was analyzed as by [31]. Nitrogen was analyzed as by [32]. Phosphorus was analyzed as by [33], and potassium was analyzed as by [34].

### 2.3. Statistical Analysis

Direct enumerated numbers of earthworms were input for calculation of density per $m^{-2}$. The biomass of earthworms per $m^{-2}$ was calculated from weight measurement of preserved wet earthworms by a digital measuring balance. The mean $\pm$ SD of density, mean $\pm$ SD biomass, and relative density of earthworms were computed. The density and biomass of earthworms per $m^{-2}$ were calculated by converting the area of one monolith into $m^{-2}$. The area of a monolith ($25 \times 25$ cm = 625 $cm^{-2}$) was converted into 1 $m^{-2}$ (10,000 $cm^{-2}$). The conversion factor (16) was determined by dividing 10,000 $cm^{-2}$ to 625. In one sampling site, there were three plots and from each plot five monoliths were dug up to extract the samples, making a total of 15 monoliths. To calculate density or biomass of earthworms per $m^{-2}$, the actual counts or measurements of earthworms were multiplied by 16 (conversion factor) and divided to 15 (number of monoliths in one site). The relative density was determined by dividing density (ind·$m^{-2}$) of each species to the total density of species in each site and multiplied by one hundred. Stata version 14 statistical software was applied to analyze variations in soil nutrient data (C, N, P, and K) between land use systems and seasons by two-way ANOVA.

Index of species richness was estimated by the Margalef species richness index [35].

$$\text{R} = \frac{S-1}{InN},$$

where R is species richness, $S$ is total number of species, $In$ is natural logarithm of number of species, and $N$ is total number of individuals in the sample.

The index of evenness was quantified by the Pielou evenness index formula [36].

$$\text{J}' = \frac{\text{H}'}{Ins},$$

where J$'$ is evenness index, H$'$ is Shannon diversity index, $s$ is total number of species in the sample, and $In$ is natural logarithm of number of species.

Species diversity was calculated by the Shannon–Wiener index [37]

$$\text{H}' = -\sum\nolimits_{i=1}^{S} \text{pi} * In\text{pi},$$

where H$'$ is Shannon diversity index, pi is relative proportion of species, and $In$ is natural logarithm of the number of species.

The index of dominance was calculated by using the Simpson diversity index formula [38]

$$D = \frac{N(N-1)}{\sum n(n-1)},$$

where $D$ is Simpson's index, $n$ is the total number of organisms of a species, and $N$ is the total number of organisms of all species.

## 3. Results

### 3.1. Physicochemical Analysis of Soil

A total of 90 soil samples were analyzed to detect the physicochemical properties of the three land use systems. The data in Table 3 revealed that the soil texture of the three land use systems fell under the same category of loam soil based on the determination of the textural triangle of the soil classification system defined by [29]. The soil temperature (°C) was greater in pasture (Pa), where there was less vegetation canopy that prevents direct sunlight from reaching the soil. The temperature values become lower in SP and

much lower in MEF. The soil moisture in Pa was lower, and inversely proportional to the soil temperatures that become higher in SP and MEF. As the decomposition process proceeded, the pH of the soil slightly increased from Pa to SP and MEF. Soil organic carbon (%), available nitrogen (kg/ha), and available phosphorus (kg/ha) also increased from Pa to SP and MEF. The available potassium (kg/ha) was much lower in Pa than in SP and MEF, but its concentration in SP was slightly greater than that in MEF (Table 3, Figures 3 and 4).

**Table 3.** Overall seasonal variation of soil physicochemical parameters in different land use types.

| Soil Parameter | Different Land Use Systems | | |
| --- | --- | --- | --- |
| | Pa Mean $\pm$ SD | SP Mean $\pm$ SD | MEF Mean $\pm$ SD |
| Sand | 22.6 $\pm$ 2.27 | 23.3 $\pm$ 2.49 | 20.3 $\pm$ 2.21 |
| Silt | 43.4 $\pm$ 3.20 | 43.2 $\pm$ 1.98 | 51.2 $\pm$ 2.04 |
| Clay | 34 $\pm$ 3.71 | 33.5 $\pm$ 3.30 | 28.5 $\pm$ 3.40 |
| Temperature (°C) | 19.91 $\pm$ 2.27 | 18.17 $\pm$ 1.72 | 16.87 $\pm$ 1.91 |
| Moisture (%) | 21.08 $\pm$ 3.59 | 25.47 $\pm$ 4.18 | 29.47 $\pm$ 5.69 |
| pH | 5.82 $\pm$ 0.12 | 6.08 $\pm$ 0.12 | 6.14 $\pm$ 0.12 |
| Organic Carbon (%) (C) | 1.378 $\pm$ 0.09 | 1.964 $\pm$ 0.12 | 2.132 $\pm$ 0.09 |
| Available Nitrogen (N) (kg/ha) | 252.53 $\pm$ 6.80 | 272.47 $\pm$ 1.64 | 272.83 $\pm$ 1.43 |
| Available Phosphorus (P) (kg/ha) | 12.36 $\pm$ 0.24 | 14.84 $\pm$ 0.18 | 15.79 $\pm$ 0.14 |
| Available Potassium (K) (kg/ha) | 89.62 $\pm$ 9.75 | 150.21 $\pm$ 4.16 | 143.25 $\pm$ 7.81 |

Pa = pasture, SP = silvopasture, MEF = mixed evergreen forest, SD = standard deviation.

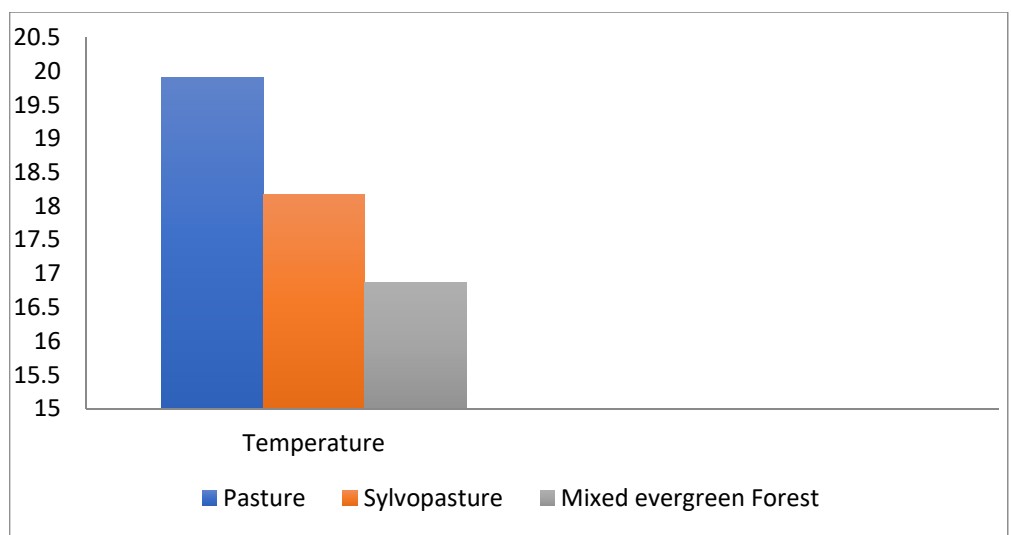

**Figure 3.** Declining of soil temperature from Pa to SP to MEF.

As indicated in Table 4, soil temperature and moisture were negatively correlated in both winter and spring (as the soil temperature value increased, the moisture values decreased). During three seasons, viz., summer, monsoon, and autumn, the values of both parameters were positively correlated as presented (as the temperature values decreased or increased, moisture values also decreased or increased). However, land use system-wise, as the temperature values increased the values of moisture decreased (Figures 3 and 4).

The two-way ANOVA output showed that the variation in organic carbon (C) concentration, potassium (K), phosphorus (P), and nitrogen (N) between land use systems were statistically significant at $p < 0.0001$. The variation in soil nutrients (C, K, and P) between seasons was also statistically significant ($p < 0.005$) except for N ($p < 0.5$) (Table 5).

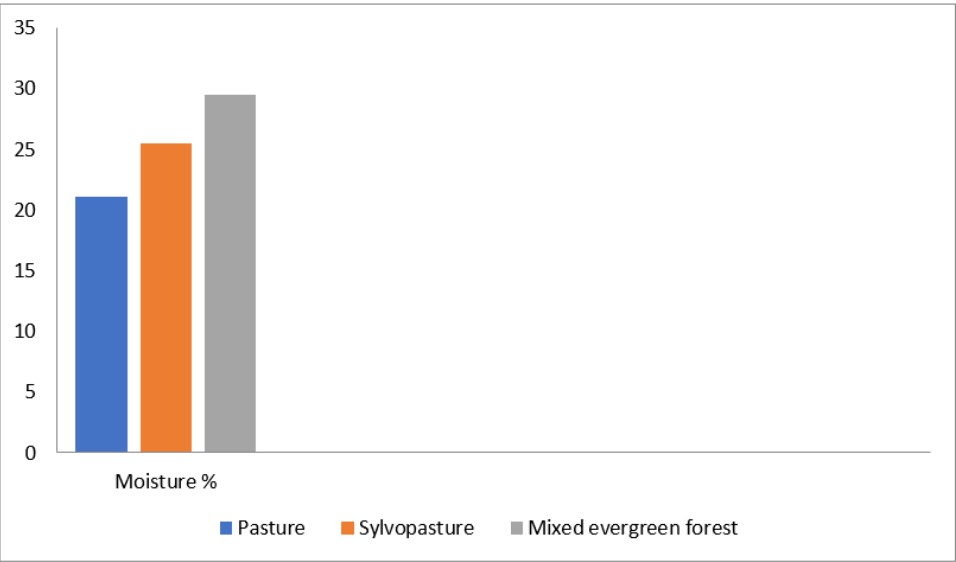

**Figure 4.** Declining of soil moisture from MEF to SP to Pa.

**Table 4.** Correlation value of temperature with moisture.

| Seasons | R |
|---|---|
| Winter | −0.4820 |
| Spring | −0.8156 |
| Summer | 0.0375 |
| Monsoon | 0.1164 |
| Autumn | 0.2946 |

**Table 5.** The two-way ANOVA result of soil nutrients (C, K, P, and N).

| Nutrients | Source | Partial SS | Df | MS | F | *p*-Value |
|---|---|---|---|---|---|---|
|   | Model | 14.167682 | 12 | 1.1806402 | 7.83 | 0.0000 |
| C | Season-wise | 3.0165 | 4 | 0.754125 | 5.00 | 0.0012 |
|   | Land use-wise | 11.151182 | 8 | 1.3938978 | 9.25 | 0.0000 |
|   | Model | 118,784.11 | 12 | 9898.6762 | 7.56 | 0.0000 |
| K | Season-wise | 19,008.19 | 4 | 4752.0474 | 3.63 | 0.0092 |
|   | Land use-wise | 99,775.925 | 8 | 12,471.991 | 9.52 | 0.0000 |
|   | Model | 204.79726 | 12 | 17.066438 | 23.94 | 0.0000 |
| P | Season-wise | 11.573649 | 4 | 2.8934122 | 4.06 | 0.0049 |
|   | Land use-wise | 193.22361 | 8 | 24.152951 | 33.88 | 0.0000 |
|   | Model | 18,030.726 | 12 | 1502.5605 | 9.59 | 0.0000 |
| N | Season-wise | 623.00875 | 4 | 155.75219 | 0.99 | 0.4160 |
|   | Land use-wise | 17,407.717 | 8 | 2175.9647 | 13.89 | 0.0000 |

The data of the concentration of nutrients (C, K, P, and N) presented in Figure 5a–d between land use systems revealed less in Pa, an increase in SP, and a great increase in MEF, respectively.

Pair-wise post hoc analyses of soil nutrients were performed to indicate the comparison of soil nutrient (C, K, P, and N) data variation between land use systems (Table 6).

The concentration of nutrients (C, K, P, and N) in different seasons indicated that the concentrations of all nutrients were greater in monsoon, except for nitrogen which was slightly greater in spring than other nutrients (Figure 6a–d).

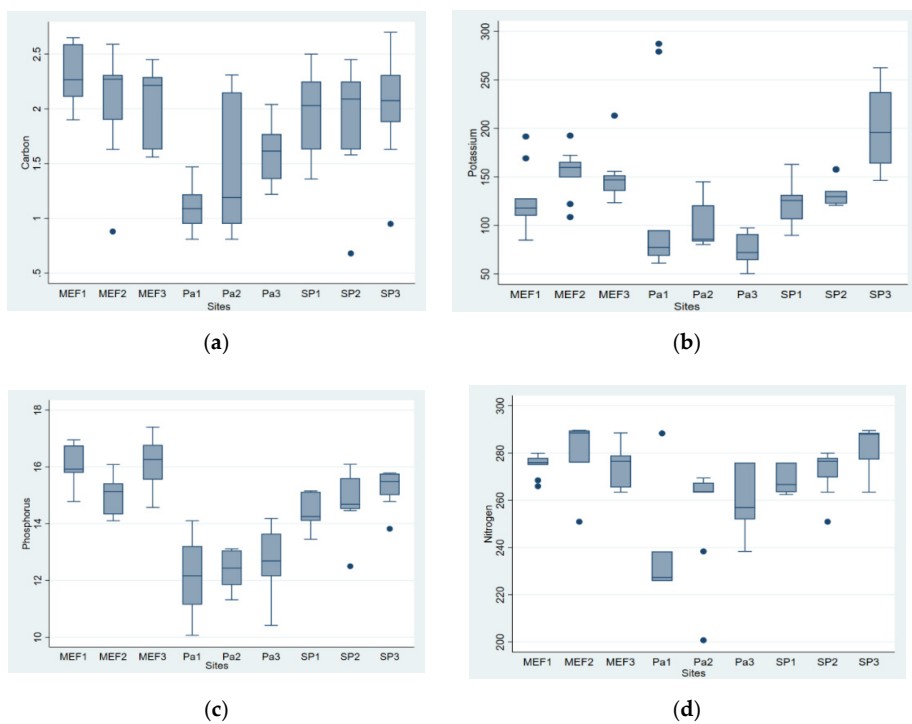

**Figure 5.** (**a**) Land use−wise carbon concentration (%), (**b**) land use-wise available potassium concentration (kg/ha), (**c**) land use-wise available phosphorus concentration (kg/ha), (**d**) land use-wise available nitrogen concentration (kg/ha); the blue dotes (outliers) in all figures are used show some anomalous results were observed in replications.

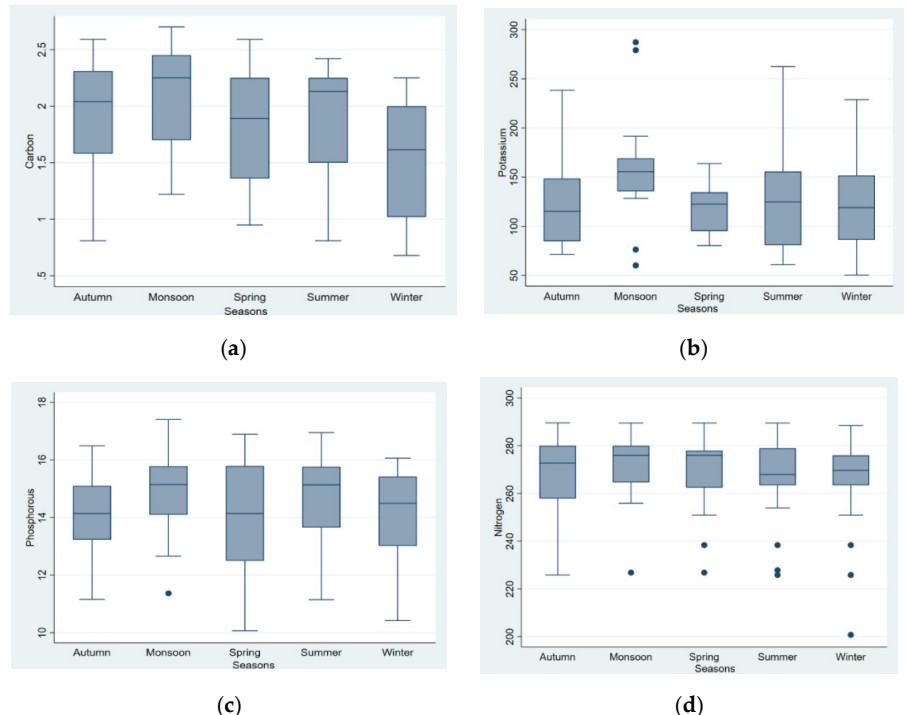

**Figure 6.** (**a**) Season-wise carbon concentration (%), (**b**) season-wise available potassium concentration (kg/ha), (**c**) season-wise available phosphorous concentration (kg/ha), (**d**) season-wise available nitrogen concentration (kg/ha); the blue dotes (outliers) in all figures are used show some anomalous results were observed in replications.

**Table 6.** Land use post hoc analysis results of soil nutrients (C, K, P, and N).

| Nutrients | Pair-Wise Comparison of Land Use Systems | Contrast | St. Error | Tukey | |
|---|---|---|---|---|---|
| | | | | t | *p*-Value |
| C | Pa1 vs. MEF1 | −1.203 | 0.189996 | −6.33 | 0.000 |
| | Pa2 vs. MEF1 | −0.836 | 0.189996 | −4.40 | 0.001 |
| | Pa1 vs. MEF2 | −0.982 | 0.189996 | −5.17 | 0.000 |
| | Pa2 vs. MEF2 | −0.615 | 0.189996 | −3.24 | 0.044 |
| | Pa1 vs. MEF3 | −0.938 | 0.189996 | −4.94 | 0.000 |
| | SP1 vs. Pa1 | 0.859 | 0.189996 | 4.52 | 0.001 |
| | SP2 vs. Pa1 | 0.818 | 0.189996 | 4.31 | 0.001 |
| | SP3 vs. Pa1 | 0.931 | 0.189996 | 4.90 | 0.001 |
| K | SP3 vs. MEF1 | 73.761 | 17.20404 | 4.29 | 0.002 |
| | Pa2 vs. MEF2 | −56.356 | 17.20404 | −3.28 | 0.039 |
| | Pa3 vs. MEF2 | −80.001 | 17.20404 | −4.65 | 0.000 |
| | Pa3 vs. MEF3 | −75.333 | 17.20404 | −4.38 | 0.001 |
| | SP3 vs. Pa1 | 81.344 | 17.20404 | 4.73 | 0.000 |
| | SP3 vs. Pa2 | 99.919 | 17.20404 | 5.81 | 0.000 |
| | SP3 vs. Pa3 | 123.564 | 17.20404 | 7.18 | 0.000 |
| | SP3 vs. SP1 | 74.255 | 17.20404 | 4.32 | 0.001 |
| | SP3 vs. SP2 | 64.879 | 17.20404 | 3.77 | 0.009 |
| P | Pa1 vs. MEF1 | −3.897 | 0.4051041 | −9.62 | 0.000 |
| | Pa2 vs. MEF1 | −3.618 | 0.4051041 | −8.93 | 0.000 |
| | Pa3 vs. MEF1 | −3.347 | 0.4051041 | −8.26 | 0.000 |
| | SP1 vs. MEF1 | −1.563 | 0.4051041 | −3.86 | 0.007 |
| | Pa1 vs. MEF2 | −2.902 | 0.4051041 | −7.16 | 0.000 |
| | Pa2 vs. MEF2 | −2.623 | 0.4051041 | −6.47 | 0.000 |
| | Pa3 vs. MEF2 | −2.352 | 0.4051041 | −5.81 | 0.000 |
| | Pa1 vs. MEF3 | −4.051 | 0.4051041 | −10.00 | 0.000 |
| | Pa2 vs. MEF3 | −3.772 | 0.4051041 | −9.31 | 0.000 |
| | Pa3 vs. MEF3 | −3.501 | 0.4051041 | −8.64 | 0.000 |
| | SP1 vs. MEF3 | −1.717 | 0.4051041 | −4.24 | 0.002 |
| | SP1 vs. Pa1 | 2.334 | 0.4051041 | 5.76 | 0.000 |
| | SP2 vs. Pa1 | 2.739 | 0.4051041 | 6.76 | 0.000 |
| | SP3 vs. Pa1 | 3.182 | 0.4051041 | 7.85 | 0.000 |
| | SP1 vs. Pa2 | 2.055 | 0.4051041 | 5.07 | 0.000 |
| | SP2 vs. Pa2 | 2.46 | 0.4051041 | 6.07 | 0.000 |
| | SP3 vs. Pa2 | 2.903 | 0.4051041 | 7.17 | 0.000 |
| | SP1 vs. Pa3 | 1.784 | 0.4051041 | 4.40 | 0.001 |
| | SP2 vs. Pa3 | 2.189 | 0.4051041 | 5.40 | 0.000 |
| | SP3 vs. Pa3 | 2.632 | 0.4051041 | 6.50 | 0.000 |
| N | Pa1 vs. MEF1 | −38.79 | 5.596977 | −6.93 | 0.000 |
| | Pa1 vs. MEF2 | −45.736 | 5.596977 | −8.17 | 0.000 |
| | Pa2 vs. MEF2 | −25.783 | 5.596977 | −4.61 | 0.000 |
| | Pa3 vs. MEF2 | −21.937 | 5.596977 | −3.92 | 0.006 |
| | Pa1 vs. MEF3 | −37.81 | 5.596977 | −6.76 | 0.000 |
| | Pa2 vs. Pa1 | 19.953 | 5.596977 | 3.56 | 0.017 |
| | Pa3 vs. Pa1 | 23.799 | 5.596977 | 4.25 | 0.002 |
| | SP1 vs. Pa1 | 32.126 | 5.596977 | 5.74 | 0.000 |
| | SP2 vs. Pa1 | 36.589 | 5.596977 | 6.54 | 0.000 |
| | SP3 vs. Pa1 | 46.809 | 5.596977 | 8.36 | 0.000 |
| | SP3 vs. Pa2 | 26.856 | 5.596977 | 4.80 | 0.000 |
| | SP3 vs. Pa3 | 23.01 | 5.596977 | 4.11 | 0.003 |

*3.2. Key Used for the Identification of Earthworms*

The preserved specimens of earthworms were identified following the key provided by [21–28].

1. Genital markings present, spermathecal atrium pear or finger—*Drawida japonica.*

2. Pale- or light pink-colored worms, calciferous sacs present in segment 10—*Aporrectodea rosea*.
3. Worms that are red-colored with yellow transverse stripes, calciferous sacs absent in segment 10—*Esenia fetida*.
4. Ventral most chaetae on segment 8 not enlarged—*Lennogaster pusillus*.
5. Spermathecal pores in intersegmental furrows 5/6/7/8; genital markings absent—*Metaphire birmanica*.
6. Spermathecal pores in furrows 6/7/8/9; genital markings small within copulatory pouches and spermathecal pore invaginations, recognizable internally by the presence of stalked glands—*Metaphire houlleti*.
7. Spermathecal pores, four pairs, at intersegmental furrows 5/6/7/8/9—*Amynthas corticis*.

### 3.3. Density and Biomass of Earthworms

Generally, the mean density of earthworms during the study period of 2019–2020 in Pa, with a value of 77.87 ind·m$^{-2}$, was lower than mean earthworm density in the other two land use systems. In SP, the mean density, with a value of 306.13 ind·m$^{-2}$, was much higher than the mean density of Pa, and the highest density was observed in MEF, with a value of 754.15 ind·m$^{-2}$. Season-wise, the highest mean density (ind·m$^{-2}$) of earthworms was generally observed in MEF, except in winter. The lowest mean density (ind·m$^{-2}$) was observed in Pa in all seasons. During winter, the mean density of earthworms was highest in SP, with a value of 87.47 ind·m$^{-2}$ (Tables 7 and 8).

**Table 7.** Density (ind·m$^{-2}$) and biomass (g m$^{-2}$) of earthworms across different land use types.

| Season and Land Use System | 2019–2020 Mean Density (Ind·m$^{-2}$) $\pm$ SE | 2019–2020 Mean Biomass (g m$^{-2}$) $\pm$ SE |
|---|---|---|
| Winter | | |
| Pa | 6.4 $\pm$ 0.16 | 4.5 $\pm$ 0.11 |
| SP | 87.47 $\pm$ 1.05 | 20.63 $\pm$ 0.29 |
| MEF | 64 $\pm$ 0.85 | 24.7 $\pm$ 0.27 |
| Spring | | |
| Pa | 35.2 $\pm$ 0.47 | 12.98 $\pm$ 0.24 |
| SP | 60.8 $\pm$ 1.38 | 16.75 $\pm$ 0.22 |
| MEF | 120.54 $\pm$ 0.50 | 20.08 $\pm$ 0.28 |
| Summer | | |
| Pa | 3.2 $\pm$ 0.14 | 1.54 $\pm$ 0.05 |
| SP | 37.32 $\pm$ 0.44 | 24.85 $\pm$ 0.39 |
| MEF | 157.87 $\pm$ 1.76 | 35.75 $\pm$ 0.31 |
| Monsoon | | |
| Pa | 22.4 $\pm$ 0.46 | 3.93 $\pm$ 0.11 |
| SP | 68.27 $\pm$ 0.94 | 36.91 $\pm$ 0.56 |
| MEF | 230.4 $\pm$ 2.38 | 43.06 $\pm$ 0.51 |
| Autumn | | |
| Pa | 10.67 $\pm$ 0.41 | 8.87 $\pm$ 0.21 |
| SP | 52.27 $\pm$ 0.52 | 25.7 $\pm$ 0.23 |
| MEF | 181.34 $\pm$ 2.75 | 32.43 $\pm$ 0.20 |
| Total | 1138.15 ind·m$^{-2}$ | 312.68 g m$^{-2}$ |

The total mean biomass of earthworms was similarly lowest in Pa, with a value of 31.82 g m$^{-2}$, but in SP, the total mean biomass was much higher than that in Pa, with a value of 124.84 g m$^{-2}$, and the total mean biomass of MEF was higher than the total mean biomass of SP, with a value of 156.02 g m$^{-2}$ (Tables 7 and 8).

**Table 8.** Biomass (g m$^{-2}$) and density (ind·m$^{-2}$) of different earthworm species in different land use types.

| Earthworm Species | Pa Ind·m$^{-2}$ Mean ± SE | g m$^{-2}$ Mean ± SE | SP Ind·m$^{-2}$ Mean ± SE | g m$^{-2}$ Mean ± SE | MEF Ind·m$^{-2}$ Mean ± SE | g m$^{-2}$ Mean ± SE |
|---|---|---|---|---|---|---|
| *A. corticis* | 45.87 ± 0.29 | 24.65 ± 0.33 | 217.6 ± 2.59 | 102.67 ± 0.80 | 109.87 ± 1.37 | 37.36 ± 0.90 |
| *A. rosea* | 8.53 ± 0.23 | 1.66 ± 0.07 | 6.4 ± 0.19 | 0.92 ± 0.02 | 605.87 ± 7.44 | 101.33 ± 2.18 |
| *D. japonica* | 12.8 ± 0.27 | 2.03 ± 0.08 | 20.27 ± 0.34 | 9.84 ± 0.39 | 20.27 ± 0.58 | 11.69 ± 0.20 |
| *M. houlleti* | 1.07 ± 0.04 | 0.51 ± 0.02 | 43.73 ± 0.37 | 7.83 ± 0.27 | 12.8 ± 0.32 | 4.4 ± 0.15 |
| *M. birmanica* | 6.4 ± 0.155 | 2.24 ± 0.09 | 8.53 ± 0.24 | 3.01 ± 0.10 | 1.07 ± 0.06 | 0.17 ± 0.01 |
| *E. fetida* | - | - | 9.6 ± 0.24 | 0.57 ± 0.02 | 4.27 ± 0.15 | 1.07 ± 0.06 |
| *L. pusillus* | 3.2 ± 0.09 | 0.73 ± 0.02 | - | - | - | - |
| Total | 77.87 | 31.82 | 306.13 | 124.84 | 754.15 | 156.02 |

### 3.4. Species-Wise Biomass and Density Status in Different Land Use Systems

Both mean biomass (g m$^{-2}$) and mean density (ind·m$^{-2}$) values in MEF were higher than those in Pa and SP. In Pa and SP, *A. corticis* was denser, and its biomass was much greater than that of the other species. However, in MEF, *A. rosea* was the densest and most abundantly occurring earthworm species. In the present study, *E. fetida* was absent in Pa, and *L. pusillus* occurred only in Pa and was absent in both SP and MEF (Table 8).

### 3.5. Species-Wise Density and Relative Density Status in Different Land Use Systems

The most dominant earthworm species in Pa was *A. corticis*, with a value of 45.87 ind·m$^{-2}$ and RD of 58.91%. *D. japonica* was the next most abundant in Pa, with a value of 12.8 ind·m$^{-2}$ and RD of 16.44%, followed by *M. birmanica* and *L. pusillus,* with abundance values and RD of 6.4 ind·m$^{-2}$ and 8.22% and 3.2 ind·m$^{-2}$ and 4.11%, respectively. *M. houlleti* occurred with 1.07 ind·m$^{-2}$ and RD of 1.37%. *E. fetida* was totally absent in Pa (Table 9).

**Table 9.** The density (ind·m$^{-2}$) and relative density (RD%) of earthworm species in different land use types.

| Earthworm Family/Species | | Ecological Category | Pa Ind·m$^{-2}$ | RD% | SP Ind·m$^{-2}$ | RD% | MEF Ind·m$^{-2}$ | RD% |
|---|---|---|---|---|---|---|---|---|
| Megascolecidae | *A. cortices* | *Epi-endogeic* | 45.87 | 58.91 | 217.6 | 71.08 | 109.87 | 14.57 |
| Lumbricidae | *A. rosea* | *Endogeic* | 8.53 | 10.95 | 6.4 | 2.09 | 605.87 | 80.34 |
| Moniligasteridae | *D. japonica* | *Endogeic* | 12.8 | 16.44 | 20.27 | 6.62 | 20.27 | 2.69 |
| Megascolecidae | *M. houlleti* | *Epi-endogeic* | 1.07 | 1.37 | 43.73 | 14.28 | 12.8 | 1.69 |
| Megascolecidae | *M. birmanica* | *Endogeic* | 6.4 | 8.22 | 8.53 | 2.79 | 1.07 | 0.14 |
| Lumbricidae | *E. fetida* | *Epigeic* | - | - | 9.6 | 3.14 | 4.27 | 0.57 |
| Octochaetidae | *L. pusillus* | *Endogeic* | 3.2 | 4.11 | - | | - | |
| Total | | | 77.87 | 100 | 306.13 | 100 | 754.15 | 100 |

In SP, *A. corticis* was again the most abundant, with a density of 217.6 ind·m$^{-2}$ and RD of 71.08%. The second most abundantly registered earthworm species was *M. houlleti*, with a density of 43.73 ind·m$^{-2}$, followed by *D. japonica* with a density of 20.27 ind·m$^{-2}$ and RD of 6.62% and *E. fetida* with a density of 9.6 ind·m$^{-2}$ and RD of 3.14%. *M. birmanica* occurred in SP, with an abundance value of 8.53 ind·m$^{-2}$ and RD of 2.79%, followed by *A. rosea*, with an abundance value of 6.4 ind·m$^{-2}$ and RD of 2.09%, but *L. pusillus* did not occur at all in SP (Table 9).

In MEF, the most abundantly occurring earthworm species was *A. rosea*, with an abundance value of 605.87 ind·m$^{-2}$ and RD of 80.34%. This earthworm species possesses a higher degree of adaptation to survive and reproduce in places where high humidity is available in the deep soil of MEF, as it is an endogeic species. The next most abundant earthworm species in MEF was *A. corticis*, with a density of 109.87 ind·m$^{-2}$ and RD of

14.57%, followed by *D. japonica*, with an abundance value of 20.27 ind·m$^{-2}$ and RD of 2.69%. The next most abundantly observed earthworm species was *M. houlleti*, with a density value of 12.8 ind·m$^{-2}$ and RD of 1.69%, followed by *E. fetida*, with a value of 4.27 ind·m$^{-2}$ and RD of 0.57%. *M. birmanica* was the least abundantly recorded species, with a value of 1.07 ind·m$^{-2}$ and RD of 0.14% (Table 9).

### 3.6. Diversity Index and Abundance of Earthworm Species in Land Use Systems

The diversity indices were calculated for Pa, SP, and MEF to evaluate the diversity status of the three land use systems (Table 10). According to the output obtained, the lowest value of the Shannon diversity index was registered in MEF at 0.661 and in SP at 0.99, and the highest value was registered in Pa at 1.25. The lowest index of evenness was registered in MEF at 0.369, then higher in SP at 0.552, and the value was the highest in Pa at 0.696. The highest value of the Margalef richness index was registered in Pa (1.165), a lower value in SP (0.883), and the lowest value in MEF (0.762). Similarly, the dominance index was also computed for the three land use systems, and the highest value was registered in MEF at 0.67, then lower in SP with a value of 0.53, and the lowest value (0.39) was registered in Pa.

**Table 10.** Diversity index of earthworms in the three land use systems.

| Earthworm Species and Parameter of Alpha Diversity | Different Land Use Systems | | |
|---|---|---|---|
| | Pa | SP | MEF |
| *A. cortices* | 43 | 204 | 103 |
| *A. rosea* | 8 | 6 | 568 |
| *D. japonica* | 12 | 19 | 19 |
| *M. houlleti* | 1 | 41 | 12 |
| *M. birmanica* | 6 | 8 | 1 |
| *E. fetida* | - | 9 | 4 |
| *L. pusillus* | 3 | - | - |
| Total number of individuals | 73 | 287 | 707 |
| Shannon index of diversity (H′) | 1.25 | 0.99 | 0.661 |
| Index of evenness (J′) | 0.696 | 0.552 | 0.369 |
| Margalef species richness index (R) | 1.165 | 0.883 | 0.762 |
| Index of dominance (D) | 0.39 | 0.53 | 0.67 |
| Species richness (n) | 6 | 6 | 6 |

### 3.7. Synchrony of Density and Biomass across Land Use Systems

The densities of earthworms were related to the vegetation of the land use systems. As its name indicates, the Pa land use system possesses less dense aboveground vegetation of grass species such as *Chrysopogon fulvus*, *Heteropogon contortus*, and *Apluda* spp. with a little litter accumulation. These vegetation types do not have enough canopy to protect the soil from sunlight. Both temperature and moisture of soil play a vital role in the survival and reproduction of soil organisms. Direct sunlight allows evaporation of water from the soil and results in less moisture content, making land use systems less suitable for the survival and reproduction of earthworms. Hence, less moisture and high temperature in Pa resulted in lower density and biomass of earthworms (77.87 ind·m$^{-2}$, 31.82 g m$^{-2}$) (Table 8).

In SP, tree species such as *Quercus leucotricophora* (Ban Oak), shrubs, and grasses were available which provide more canopy to prevent direct sunlight from reaching the soil than in Pa. The temperature and moisture contents in SP were more suitable for survival and reproduction and hence the density and biomass of earthworms were greater (306.13 ind·m$^{-2}$, 124.84 g m$^{-2}$) (Table 8). MEF possesses more vegetation such as *Quercus leucotricophora* (Ban Oak), *Pinus roxburghii* (Chir Pine), *Myrica nagi* (Kafal), *Prunus paddus* (Paja), *Celtis australis* (Khirak), *Toona ciliata* (Toon), *Ficus* spp. (Anjeer), *Indigofera pulchella* (Kathi), *Beberis asiatica* (Rasaunt/Kasgmal), *Rubus ellipticus* (Yellow Himalayan Raspberry), and *Carrisa coronda* (Coronda) with much accumulation of litter. The vegetation provides more survival and reproductive advantages for earthworms and hence their density and biomass increased to 754.15 in.m$^{-2}$, 156.02 g m$^{-2}$ (Table 8).

### 3.8. Relationship of Density and Biomass with Diversity

The data of density and biomass of earthworms were negatively correlated with diversity values (Table 11). According to the recorded data in Pa (77.87 ind·m$^{-2}$ and 31.82 g m$^{-2}$), in SP (306.13 ind·m$^{-2}$ and 124.84 g m$^{-2}$), and in MEF (754.15 ind·m$^{-2}$ and 156.02 g m$^{-2}$), both density and biomass (productivities) of earthworms increased from Pa to MEF with the increasing aboveground vegetation (MEF > SP > Pa). However, the diversity index values of Shannon (H'), evenness (J'), and Margalef richness indices (R) increased in Pa and decreased in SP and were the lowest in MEF. The values decreased from Pa to MEF as indicated in Table 9, from Pa (H' = 1.25, R = 1.165, J' = 0.696), to SP (H' = 0.99, R = 0.883, J' = 0.552), to MEF (H' = 0.661, R = 0.762, J' = 0.369) (Pa > SP > MEF). On the other hand, the highest dominance index (D) value was registered in MEF (Pa = 0.39, SP = 0.53, MEF = 0.67). The values of alpha diversity revealed that in the places where there was less vegetation, in, e.g., Pa, high belowground diversity occurred and in the places where more vegetation was observed, e.g., in MEF, a lower diversity of earthworms was recorded.

**Table 11.** The relationship between diversity, density, and biomass of earthworms.

| Population Parameter | Land Use System | | |
| --- | --- | --- | --- |
| | **Pa** | **SP** | **MEF** |
| Density (ind·m$^{-2}$) | 77.87 | 306.13 | 754.15 |
| Biomass (g m$^{-2}$) | 31.82 | 124.84 | 156.02 |
| Diversity parameter | | | |
| Shannon index of diversity (H') | 1.25 | 0.99 | 0.661 |
| Index of evenness (J') | 0.696 | 0.552 | 0.369 |
| Margalef species richness index (R) | 1.165 | 0.883 | 0.762 |
| Index of dominance (D) | 0.39 | 0.53 | 0.67 |
| Species richness (n) | 6 | 6 | 6 |

## 4. Discussion

### 4.1. Physicochemical Analysis Results of Three Different Land Use Systems

The physicochemical parameters of soil are independent variables that affect the diversity, density, and biomass (dependent variables) of earthworms. The soil physicochemical parameters of different land use systems (Pa, SP, and MEF) are influenced by the absence and availability of aboveground vegetation [39]. There was an investigation that evaluated how changes in land use systems such as grassland, plantation forest, and natural forest influenced the decrement and increment of soil nutrients in Ethiopia and reported that in natural forest, the concentration of soil nutrients was greatest as compared to other land use systems [40]. The change from natural forest to farmland and to other types decreased soil nutrients such as organic carbon, total nitrogen, and available phosphorus [11]. Another study also reported that conversion of land use systems influenced soil nutrient dynamics in Amazonia [9] and other research revealed changes in land use systems able to affect soil physicochemical parameters [11]. According to the results presented in Table 4, the recorded mean ± SD of temperature in Pa (19.91 ± 2.27 °C) was the highest, lower in SP (18.17 ± 1.72 °C), and the lowest in MEF (16.87 ± 1.91 °C). As the temperature increases, the moisture decreases, and temperature influenced the percentage of moisture in the soil (Figures 3 and 4). The Pa land use system possessed a lower moisture content (21.08 ± 3.59%), as a higher temperature allowed loss of water from the soil through evaporation. The moisture content of the soil in the SP land use system increased (25.47 ± 4.18%) and it was the highest in MEF (29.47 ± 5.69%) (Table 3, Figures 3 and 4).

It was reported that the functional composition and species diversity of plants (above ground) affected the concentration of organic carbon, available nitrogen, and phosphorus in the soil [41]. Accordingly, the concentrations of organic carbon (%), available nitrogen (kg/ha), available phosphorus (kg/ha), and available potassium (kg/ha) in Pa were lower than those in SP and MEF. In Pa, there was less dense aboveground vegetation, resulting in higher temperature (°C) and lower moisture contents (%), which in turn affected the

decomposition of organic matter and the availability of these nutrients. In this study, except for available potassium (K) (kg/ha), all soil nutrients (C, N, P) analyzed were highest in MEF. The available potassium (kg/ha) of SP was much greater than that of Pa and slightly greater than that of MEF (Table 3). The data in Figure 5a–d show that the concentrations of nutrients (C, K, P, and N) were lower in Pa, greater in SP, and the greatest in MEF, respectively. The variations in nutrients between land use systems were statistically significant as presented in Table 5 and with a pair-wise post hoc test (Table 6).

### 4.2. Density and Biomass of Earthworms

The density (ind·m$^{-2}$), relative density (RD%), and biomass (g m$^{-2}$) of all earthworms are presented in Tables 7–9. The MEF comprised tree species such as *Quercus leucotricophora* (Ban Oak), *Pinus roxburghii* (Chir Pine) *Myrica nagi* (Kafal), *Prunus paddus* (Paja), *Celtis australis* (Khirak), *Toona ciliata* (Toon), and *Ficus* spp. (Anjeer). The shrub species were *Indigofera pulchella* (Kathi), *Beberisa sciatica* (Rasaunt/Kashmal), *Rubus ellipticus* (Hinr), and *Carrisa coronda* (Coronda) with much accumulation of leaf litter. Higher densities of these tree and shrub species allowed a higher density of earthworms to occur in MEF. It was found that changes in land use systems resulted in changes in the earthworm community [42]. Earthworm density was mainly influenced by aboveground vegetation [43], which confirmed that the aboveground vegetation supported belowground organisms and vice versa.

The higher density of aboveground vegetation directly influenced the higher density and biomass of belowground organisms by supplying food resources and stabilizing the physicochemical properties of the soil. Canopies of MEF serve to provide shade and conserve moisture in the soil, stabilize temperature, and facilitate the breakdown of organic matter, which supports the survival of belowground organisms [44]. The canopies protect the soil from direct sunlight, creating conditions that are conducive to the survival and reproduction of belowground organisms [13]. Accordingly, the highest density of earthworms was observed in MEF, where a higher density of vegetation was observed as compared to other land use systems (Pa and SP). In addition, earthworm species in the three land use systems (*A. rosea*, *M. houlleti*, *M. birmanica*, *D. japonica*, and *L. pusillus*) were endogeic. *E. fetida* is epigeic and *A. corticis* is epi-endogeic [45]. Therefore, MEF was the preferred habitat for most species of the study sites. The effect of land use systems having large amounts of aboveground vegetation was confirmed by the data presented in Table 8. The highest mean density (754.13 ind·m$^{-2}$) of earthworms was recorded in MEF, followed by SP with a value of 306.13 ind·m$^{-2}$, and the lowest density in Pa with a value of 77.87 ind·m$^{-2}$.

The variation in the values of the three land use systems was due to the variation in physicochemical parameters caused by the shade provided from vegetation of these different land systems and availability of resources. In the Pa land use system, the vegetation was limited to grass species such as *Chrysopogon fulvus*, *Heteropogon contortus*, and *Apluda* spp., with less litter accumulation, and therefore, the land use system was limited in resources, which resulted in limited density.

As presented in Table 8, the mean biomass of earthworms was lowest, with a value of 31.82 g m$^{-2}$ in Pa compared to the other land use systems (SP with a value of 124.84 g m$^{-2}$ and MEF with a value of 156.02 g m$^{-2}$), as in the case of the mean density. The Pa land use type was devoid of resources due to the lower density of aboveground vegetation, resulting in physicochemical conditions being less suitable for the survival and reproduction of earthworms. SP was more suitable than Pa and subsequently resulted in more biomass. Most earthworm species, such as *A. rosea*, *D. japonica*, *M. houlletii*, and *M. birmanica* (endogeic), were more adapted to survive in a soil habitat where large amounts of leaf litter were deposited in MEF. The result revealed that MEF was the most suitable land use system for the survival and reproduction of the worms and this caused the biomass to increase. The values of density and biomass of earthworms were high in monsoon (Table 7) and were directly related to the highest values of soil nutrients (Figure 6a–d). Monsoon is the rainy season of north India characterized by highest moisture contents of soil that allow

decomposition and availability of soil nutrients to support the survival and reproduction of earthworms. The lowest value of soil nutrients was generally recorded in Pa, while it was greater in SP and the greatest in MEF (Table 3, Figure 5a–d). These values directly influenced the survival and reproduction of earthworms and subsequently affected density and biomass.

According to one study, overall density and biomass of earthworms in different land use systems were determined by plant diversity and soil properties. They divided hilly land use types into annual dominated (AD), mixed (MX), and perennial dominated (PD) and investigated how aboveground and soil properties affected density and biomass of earthworms in the USA and found that both biomass and density of earthworms in mixed (MX) land use were significantly greater than the two land use systems in which plant diversity was the greatest [46]. Greater diversity of vegetation facilitates soil physicochemical properties, conducive for survival and reproduction of earthworms. In addition, [8] investigated the abundance of earthworms in different land use systems such as coniferous forests, intensively managed agricultural land use, organic no-till systems, and ancient deciduous forest systems. The result revealed that the highest population of earthworms was recorded in ancient deciduous forest systems.

### 4.3. Diversity of Earthworm Species

A total of seven species of earthworms were identified in the three land use systems. In both Pa and SP, *A. corticis* (epi-endogeic) was dominant, but in MEF, *A. rosea* (endogeic) was the dominant species (Table 10). Adaptation (pigmentation) helps *A. corticis* to survive in sunlight-exposed habitat, and it occurred more abundantly in Pa than other species [47]; again, due to its adaptation to dwell in the soil epi-endogeically, *A. corticis* occurred more abundantly in MEF alongside *A. rosea*. *A. rosea* is exclusively endogeic and best adapted to survive in forests where leaf litter accumulates, and rarely occurred in Pa.

Diversity indices were computed for the three land use types separately to compare the diversity status (Table 8). Lower values of the Shannon diversity index (H′), Margalef species richness index (R), and evenness index (J′) imply a lower diversity in the given area. Among the three land use systems, the Shannon diversity index value was lowest in MEF, higher in SP, and the highest in Pa, with values of 0.661, 0.99, and 1.25, respectively, indicating that MEF was the least diversified land use system. Evenness (J′) was also applied, and the lowest evenness value was registered in MEF, a higher value in SP, and the highest value in Pa of 0.369, 0.552, and 0.696, respectively. The lowest value of evenness also indicated the lowest diversity. Another diversity parameter value was also computed to determine species richness by the Margalef richness index (R) with its highest value in Pa (1.165), then SP (0.883) and MEF (0.762). Based on these diversity index values, MEF was the least diversified land use system. In contrast, the greatest Simpson dominance index was registered in MEF with a value of 0.67, while it was lower in SP with 0.53, and lowest in Pa with a value of 0.39 (MEF > SP > Pa), implying again that MEF was less diversified. The diversity values implied that MEF was dominated by one or two species of earthworms that developed adaptations to dwell in the soil epi-endogeically (*A. corticis*) and endogeically (*A. rosea*). The study revealed that the dominance of certain species of earthworms in a land use system decreased the diversity index value and made the area less diversified. The area which was dominated by *Metaphire tschiliensis* greatly reduced the diversity status of the land use system [48].

This study confirmed that the diversity of earthworms was not positively correlated with density of vegetation of land use systems (Pa, SP, and MEF). The density and biomass of earthworms were much lower in the land use system with less dense vegetation, while they increased as the density of vegetation increased [49]. In the three land use systems, the species richness (n) was the same, which matched with the work of [7] which reported that changes in land use systems do not necessarily cause a change in species richness in the same landscape. A study investigated the population status of earthworms in different land use systems and found that the change from the traditional pure crop system to the

traditional agroforestry system significantly increased earthworm density and biomass but not species richness in a landscape of the central Himalayas, India [7]. Another study [15] investigated the effect of land use system conversion and reported that density and biomass of earthworms increased in aged farm land, while species diversity, richness, and evenness decreased. According to this study, the farmland was protected from human activities and became more suitable for the survival of certain species of earthworms only, causing the land to be dominated by a few species and minimizing its diversity.

*4.4. Trends of Productivity (Density and Biomass) and Diversity across Land Use Systems*

Alpha diversity and productivity (biomass and density) values of earthworms in different land use systems revealed that factors that influence earthworm diversity differ from factors that influence their density and biomass. According to the result of this study, MEF was dominated by two species of earthworms (*A. corticis* and *A. rosea*) which made the area less diversified, implying dominancy reduces diversity. The alpha diversity index value indicated that the Pa land use system was the most diversified, SP was less diversified, and MEF was the least diversified. Both earthworm species (*A. corticis* and *A. rosea*) developed adaptations to survive in the soil epi-endogeically and endogeically, respectively, and were able to survive and reproduce better than other earthworm species in MEF. They possess very little or no cutaneous pigment to survive in less intense light and are able to feed directly on soil materials. These adaptations enable them to survive in the soil and become more dominant in the place where many litter layers accumulated and made MEF less diversified (Table 11).

The productivity (biomass and density) values of earthworms in the three land use systems were the lowest in the Pa land use system, greater in SP, and greatest in MEF (Tables 8 and 9). In the Pa land use system, the soil temperature value was the highest which does not allow moderate moisture due to a smaller canopy of vegetation. In SP, more vegetation was available to provide a canopy and prevented direct sunlight from reaching the soil, resulting in a more suitable temperature and moisture than in Pa. MEF was the most suitable habitat for the survival and reproduction of most earthworms and hence had more productivity [47]. According to the finding of this study, the trend of productivity decreased from MEF to Pa (MEF > SP > Pa) (Table 8) and diversity increased from MEF to Pa (Pa > SP > MEF) (Table 11). Therefore, temperature and moisture contents of land use systems are the driving factors behind determination of biomass and density of earthworms.

**5. Conclusions**

The density, diversity, and biomass of earthworms were studied over a period of two years from 2019–2020 in Solan district, Himachal Pradesh, India, in three different land use systems, Pa, SP, and MEF, to evaluate how the change in land use systems impacted the population status of earthworms. The lowest values of H′, R, and J′ were registered in MEF, indicating that it was the least diversified land system. In contrast, the highest value of the dominance index (D) in MEF also revealed that MEF was the least diversified land use system. Among the seven earthworm species identified from the three land use systems, *A. corticis* was dominant in both Pa and SP, but *A. rosea* was the dominant species in MEF. The survival of *A. corticis* and *A. rosea* depended on the adaptation of habitat preference as epi-endogeic and endogeic, respectively. The species richness (n) in the three land use systems was the same. The mean density and mean biomass of earthworms were lower in Pa, where there was a lower density of aboveground vegetation, and the density and biomass of earthworms subsequently increased in SP and MEF, where the density of aboveground vegetation was the highest. The four measures of alpha diversity (H′, J′, R, and D) showed that MEF was the least diversified land use system, implying that the productivity (biomass and density) values were negatively correlated with the diversity values of the soils in the land use systems (productivity increased while diversity decreased). The results showed that the productivity (biomass and abundant) of earthworms was greatly influenced by

changes in land use systems and showed perfect synchrony between aboveground and belowground habitat. Therefore, it is recommended that for belowground productivity, the conservation of aboveground habitat should not be neglected and vice versa.

**Author Contributions:** K.G.: Conceptualization, Visualization, Software, Methodology, Validation, Writing—review and editing; A.B.: Visualization, Software, Methodology, Validation; J.M.J.: Visualization, Software, Methodology, Validation; S.A.B.: Visualization, Software, Methodology, Validation, Funding; G.K.S.: Visualization, Software, Methodology, Validation; G.B.: Visualization, Software, Methodology, Validation, Writing—review and editing; R.A.: Writing—review and editing; M.B.: Software, Methodology, Validation, Writing—review and editing. All authors have read and agreed to the published version of the manuscript.

**Funding:** The research received no external funding.

**Institutional Review Board Statement:** Not applicable.

**Informed Consent Statement:** Not applicable.

**Data Availability Statement:** Not applicable.

**Acknowledgments:** We thank Shoolini University for facilitating transportation during sampling in each season of two years and supplying lab facilities for soil analysis and species identification.

**Conflicts of Interest:** The authors declared no conflict of interest.

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
