# Peer review of "Impact of Aboveground Vegetation on Abundance, Diversity, and Biomass of Earthworms in Selected Land Use Systems as a Model of Synchrony between Aboveground and Belowground Habitats in Mid-Himalaya, India"

_soilsystems, doi:10.3390/soilsystems6040076_

Round 1

Reviewer 1 Report

Reviewer Comment to manuscript soilsystems-1783813

[Impact of aboveground vegetation on abundance, diversity, and biomass of earthworms in selected land use systems as a model of synchrony between aboveground and belowground in Mid Himalaya, India]

Dear Authors,

The article is publishable after clarifying the comments:

“Julka, J.M. Fauna of India and the adjacent countries. 1988 "- the bibliographic record of this key on the basis of which the species of earthworms was identified is insufficient to find it in the database. Why did the authors not use the key to identify earthworms, which is commonly used by researchers?

Were 15 samples taken on each experimental site in each research season?

Figure 3 and 4 present the data contained in Table 3.

Table 5 - the contents of the table are unreadable. Why is the table not showing whether the differences are statistically significant.

Why do the authors use a strange notation, e.g. No/m-2, g/m-2 (in table 5) and in table 6 (No/m2)? Should be: ind.·m-2, g·m-2 (ind. = individual).

Author Response

Dear reviewer 

We have made a change on manuscript based on your comments 

The change we have made are red highlighted in the manuscript also explained in cover letter. We have also edited its language by Extensive English editing service. 

Best Regards 

Dr. Sartaj Ahmad Bhat and Mr . Kasahun Gudeta  

Reviewer 2 Report

Dear authors,

You have collected great data which should be published. However, in the present form your manuscript is not doing the data and your hard field work justice.

I would like you to look at your manuscript and grammar check it. There are many instances where sentences are awkward For example, look at the sentence on line 46. It is not clear what influences what because of the construction of the sentence. The sentence is: Change of land use system is greatly influence diversity, distribution and abundances of earthworms [1, 2, 3]. Do you mean Land use change influences diversity OR do you mean Land use is greatly influenced by diversity? I assume the former is true, but then you need to take out the is.

There are many other instances of this kind of error. One other example is in lines 322-324. Take out the word "was'. Have somebody with a good sense of English look over the manuscript.

Define abbreviations at first occurrence in the main part of the manuscript. Choose a different abbreviation for the site you call P as you already using P for phosphorus.

IN the introduction, state your objectives and a hypothesis (or several) that you can test with your data. Refer back to the hypothesis in your results section and state whether you can infer (from statistics) that your hypothesis is either rejected or accepted.

I know that you have a list of earthworm species in the abstract. However, in the main part of the text you still need to name the full species name on first occurrence.

In the materials and methods section give the equations of the diversity indices you measured (also call them indices and not diversity status).

On the scientific side, I would carefully look at the statistics and at what your soil fertility measurements mean. Statistics first, you have essentially an experimental (sampling) design with two variables: site and date. This requires a two-way ANOVA (repeated measures) and the results of the analysis should be shown in an ANOVA table. Then do some sort of post-hoc test.

The fertility measurements are not clear. a. mini-kjeldahl gives total N, not available N. b. I assume you extracted P and K with Bray and then measured them.  So first in the table, don't call your N measurement 'available', put Kjeldahl instead.

You say that there is an inverse proportional relationship between temperature and moisture. You need to test that statement with a linear regression. proportional means linear but it is not clear from your graph that it is a linear relationship.

The units for earthworm abundance should be m-2. No need for the No./m-2.

MDPI journals use the numbered reference style. Don't mix that style with the Author (date) style as you do often in the manuscript.

It is not clear how you dealt with earthworm counts in your statistics. Usually one needs to use a generalized linear model with a Poisson distribution and a log-link function. Counts are not usually normally distributed (rather they follow a Poisson distribution). ANOVA does not work in cases where you have counted data.

Author Response

Dear reviewer 

We have corrected the manuscript based on the comments provided. The corrected part of the manuscript highlighted by red and also indicated and explained point by point fashion in the cover letter 

Best regards 

Dr. sartaj Ahmad Bhat and Mr. Kasahun Gudeta 

Reviewer 3 Report

Dear Authors,

Despite the work presented being interesting, the manuscript should be meliorated and rewritten.

General comment:

- the statistical approaches involved in this study are very basic and incomplete; more statistical approaches can be done by multivariate analysis (PCA, CCA).

- The results are poorly displayed. Tables and figures need to be redone: 1) table 3 does not contain the seasonal variation of soil physicochemical properties; in all tables, 2 decimal digits are sufficient for the values of SD.

-  For Figure 3 it would be better to make two graphs for temperature and moisture with two different units of measurement.

- Figure 4 is redundant because it shows the same data as in table 3; moreover, the differences in the percentages of organic carbon are not appreciable because in a unit of measurement different from that of available nitrogen, phosphorous, potassium (kg/ha).

- In table 4 the statistical analysis is unclear.

-  Some of the data in table 6 are already reported in table 5.

-  In table 5 there is no ANOVA of alpha-diversity indices.

-  The discussion reports the results again in several points and a few comparisons with other works.

-  The conclusions are obvious and report synchrony between aboveground and belowground that has not been analyzed and discussed.

I'm sorry, but I think the manuscript needs to be completed and completely rewritten

Author Response

Dear reviewer 

We have amended the manuscript based on your comments. The corrected part of the manuscript has been red highlighted and also explained in the attached cover letter point by point fashion.  The language of the manuscript  also edited by English editing service and certified for that. Kindly, check whether we have corrected or not. 

Best Regards 

Dr. Sartaj Ahmad Bhat and Mr Kasahun Gudeta 

Round 2

Reviewer 2 Report

The only thing I ask the authors do now is that they define MEF, SP, and Pa in the main part of the paper. I know they already did this in the abstract, but they have to do it one more time in the main part of the manuscript. They can do this at first appearance of the abbreviation. No need for an additional sentence. After that I am fine with publication.

Thanks for working hard on the revision of your manuscript. You did a lot of interesting work!

Author Response

Dear reviewer, we have corrected the comments provided according to the suggestion provided. All the changes we made have been highlighted yellow. Kindly check text in the manuscript. 

Best regards

Authors 

Reviewer 3 Report

Dear Authors, I appreciate the effort to improve the manuscript.

The most critical aspects have been corrected: figures and tables have been enhanced and made easier to interpret; the statistical analysis was deepened; the discussion reports more comparisons with other works with the addition of new bibliographic references.

There are still some inaccuracies to correct: figures 3 and 4 have the caption reversed and could be improved the layout; the caption of figures 5 and 6 could be improved; in table 10 the value of M. houlleti in SP is not in columns.

I believe that after these minor revisions, the manuscript has been sufficiently improved to warrant publication on Soil Systems

Author Response

Dear reviewer, we have corrected comments and suggestions provided to improve our manuscript for publication. We have also attached cover letter to report you. Kindly, we request to check the highlighted part of the manuscript.

Best regards
